# A Modified YOLOv4 Deep Learning Network for Vision-Based UAV Recognition

Farzaneh Dadrass Javan [1,2,*], Farhad Samadzadegan [2], Mehrnaz Gholamshahi [3] and Farnaz Ashatari Mahini [2]

1 Faculty of Geo-Information Science and Earth Observation (ITC), University of Twente, 7522 NB Enschede, The Netherlands

2 School of Surveying and Geospatial Engineering, College of Engineering, University of Tehran, Tehran 1439957131, Iran; samadz@ut.ac.ir (F.S.); f.ashtari@ut.ac.ir (F.A.M.)

3 Department of Electrical and Computer Engineering, Faculty of Engineering, Kharazmi University, Tehran 1571914911, Iran; mehrnazgholamshahi.khu@gmail.com

* Correspondence: f.dadrassjavan@utwente.nl

**Abstract:** The use of drones in various applications has now increased, and their popularity among the general public has increased. As a result, the possibility of their misuse and their unauthorized intrusion into important places such as airports and power plants are increasing, threatening public safety. For this reason, accurate and rapid recognition of their types is very important to prevent their misuse and the security problems caused by unauthorized access to them. Performing this operation in visible images is always associated with challenges, such as the small size of the drone, confusion with birds, the presence of hidden areas, and crowded backgrounds. In this paper, a novel and accurate technique with a change in the YOLOv4 network is presented to recognize four types of drones (multirotors, fixed-wing, helicopters, and VTOLs) and to distinguish them from birds using a set of 26,000 visible images. In this network, more precise and detailed semantic features were extracted by changing the number of convolutional layers. The performance of the basic YOLOv4 network was also evaluated on the same dataset, and the proposed model performed better than the basic network in solving the challenges. Compared to the basic YOLOv4 network, the proposed model provides better performance in solving challenges. Additionally, it can perform automated vision-based recognition with a loss of 0.58 in the training phase and 83% F1-score, 83% accuracy, 83% mean Average Precision (mAP), and 84% Intersection over Union (IoU) in the testing phase. These results represent a slight improvement of 4% in these evaluation criteria over the YOLOv4 basic model.

**Keywords:** convolutional neural network CNN; YOLO deep learning; drone; UAV; drone detection; drone recognition

## 1. Introduction

Drones are actively used in a variety of fields, including recreational, commercial, security, crisis management, and mapping [1,2]. They are also used in combination with other platforms such as satellites in resource management, agriculture, and environmental protection [3–5]. However, the negligent and the malicious use of these flying vehicles poses a great threat to public safety in sensitive areas such as government buildings, power plants, and refineries [6,7]. For this reason, it is important to recognize drones to prevent them from entering critical infrastructure or ensuring security in large locations such as stadiums [8].

In this study, the recognition of four types of drones was investigated. Conventional drone detection technologies include the use of various sensors such as radar (radio detection and ranging) [9], Lidar (Light Detection and Ranging) [10], acoustic [11], and thermal sensors [12]. In these methods, first, the presence or the absence of the drone in the scene is checked and then the drone type recognition process is performed [13,14]. However, the

application of these types of sensors has always been associated with problems such as higher costs and higher energy consumption [15]. In contrast, visible images do not have these problems and are widely used for object recognition and semantic segmentation due to their high resolution [16]. On the other hand, the use of visible images also introduces problems such as light changes within the imagery, the presence of occluded areas, and a crowded background, which necessitates the application of an efficient and comprehensive method for recognition.

Recent advances in deep convolutional neural networks and the appearance of more improved hardware make it possible to use visual information to recognize objects with higher accuracy and speed [17]. Unlike conventional drone detection technologies, the nature of deep learning networks is to perform drone recognition simultaneously. By classifying inputs into several classes, these networks determine the presence, absence, image location, and type of drone class [18]. Among neural networks, the convolutional neural network (CNN) is one of the most important representatives of image recognition and classification. In this network, the input data enters the convolutional layers. The convolution operation is then performed using the network kernel to find similarities. Finally, feature extraction is performed using the resulting feature map [19]. There are different types of convolutional neural networks available such as R-CNN (Region-based CNN) [20], SPPNet (Spatial Pyramid Pooling Network) [21], and Faster-RCNN [22]. In these networks, due to the application of convolutional operations, more features are extracted than in conventional object detection methods and better speed and accuracy are achieved in recognizing objects. The extracted features are essentially descriptors of objects, and as the number of these features increases, object recognition is performed with higher accuracy. In these networks, the proposed regions are first defined using region proposal networks (RPNs) [23]. Then, convolutional filters are applied to these regions, and the extracted features are obtained as the result of the convolutional operation [22]. In other deep learning methods such as SSD (Single Shot MultiBox Detector) [24] and YOLO [25], the image is generally explored, which results in higher accuracy and speed in object recognition as compared to the basic methods [25]. The reason for the higher speed in these methods is the architecture is simpler than in region-based methods. The YOLO network is a method for detecting and for recognizing an object based on CNNs. The YOLO network predicts bounding box coordinates and class probabilities for these boxes, considering the whole image. The fourth edition of the YOLO Network is the YOLOv4 Deep Learning Network, which performs better than previous versions in terms of speed and accuracy [26]. However, the YOLOv4 deep learning network may not be able to overcome some challenges, such as the small size of the drone in different images [16]. In this study, this network could not recognize the drone in some of the challenging images. These challenges include confusing some drones with birds due to their small size, and the presence of drones in crowded backgrounds and hidden areas. Therefore, the YOLOv4 deep learning network was modified to better overcome the challenges of recognizing flying drones. The change in the architecture of this network is the main innovation in this article. Also, 4 types of multirotors, fixed-wings, helicopters, and VTOLs (Vertical Take-Off and Landing) were recognized. Given the need to recognize each type of UAV in different applications, the study of this topic can be considered as another innovation of this paper.

### 1.1. Challenges in Drone Recognition

Drone recognition is always fraught with challenges. Some of the important challenges in this regard are discussed.

### 1.1.1. Confusion of Drones and Birds

Due to the physical characteristics of drones, they can easily be confused with birds in human eyes. This problem is more challenging when using drones in maritime areas due to the presence of more birds. The similarity between drones and birds and their distinction from each other is shown in Figure 1.

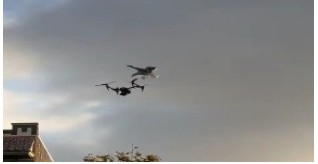 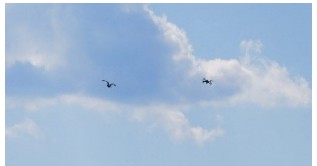 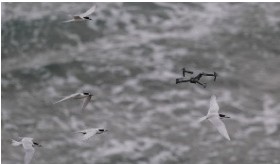

**Figure 1.** Challenges related to confusion with birds in drone recognition.

### 1.1.2. Crowded Background

As it appears from Figure 2, the presence of drones in areas with crowded backgrounds and similar environments has made them more difficult to recognize due to the inability to isolate the background. A crowded background refers to conditions such as the existence of clouds, dust, fog, and fire in the sky.

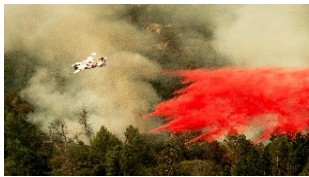 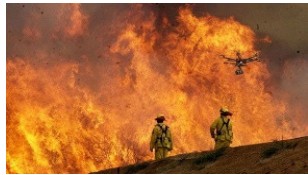 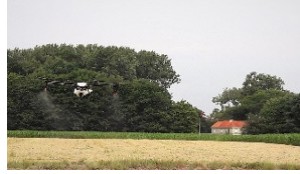

**Figure 2.** Challenges related to a crowded background in drone recognition.

### 1.1.3. Small Drone Size

The small size of drones makes them difficult to see at longer distances and difficult to quickly and accurately recognize, or they are possibly recognized as birds. Furthermore, the presence of a swarm of UAVs at different scales makes the recognition process more challenging. Figure 3 illustrates some examples of the presence of small drones at different scales.

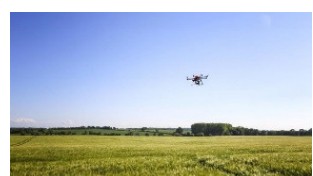 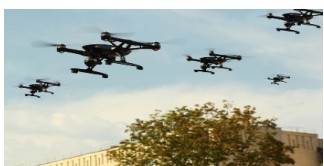 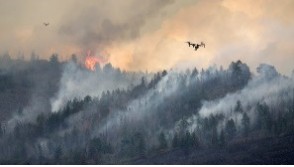

**Figure 3.** Challenges related to the presence of small drones at different scales.

Drone recognition is always fraught with challenges. For this reason, it is necessary to use a fast, accurate, robust, and efficient method to overcome the challenges and to correctly recognize drones.

## 2. State of the Art Work

Recently, the use of drones has become increasingly popular, and they have been applied to various scientific and commercial purposes in different fields of photogrammetry, surveying, agriculture, natural disaster management, and so on [27]. There are different types of drones, each used for a specific purpose; and, in terms of design technology, application, and physical characteristics, they can be divided into four types: multirotor, helicopter, VTOL, and fixed-wing [28–31]. They are also divided into two scenarios in terms of operation manner in the environment. In the first scenario, drones operate individually, while in the second scenario they fly in combination with others, which are normally known as a swarm of UAVs [32–35].

Because of the enormous potential applications of each type of UAV in meeting the needs of society, the possibility of their misuse has become a major concern for communities. Over the past decade, much of the research has focused on finding efficient and accurate

techniques for the recognition of different types of UAVs [12,17,36]. However, sometimes drone recognition is difficult because they are normally flying in challenging environments. Therefore, the recognition of UAVs requires advanced techniques that can recognize them as they fly individually or in swarm mode.

## 3. Related Works

Due to the increasing development of deep neural networks in visual applications, these networks are also used widely for the recognition of objects in visible images [36–38]. In 2019, Nalamati et al. used a collection of visible images to detect small drones and solve their detection challenges. In this work, different CNN-based architectures were used, such as SSD [22], Faster-RCNN with ResNet-101 [22], and Faster-RCNN [22] with Inceptionv2. Based on the results, the R-CNN network with ResNet-101 performs the best in training and testing [39]. In 2019, Unlu et al. used an independent drone detection system, using the YOLOv3 Deep Learning Network. One of the advantages of this system is its cost-effectiveness due to the limited need for GPU memory. This study can detect drones of a small size and at a minimal distance, but it cannot recognize the types of drones [40]. In 2020, Mahdavi et al. detected a drone using a fisheye camera, and three methods of classification were applied: convolutional neural network (CNN), support vector machine (SVM), and nearest-neighbor. The results showed that CNN, SVM, and nearest-neighbor have total accuracy of 95%, 88%, and 80%, respectively. Compared with other classifiers with the same experimental conditions, the accuracy of the convolutional neural network classifier was satisfactory. In this study, only the detection of drones without considering their types and challenges has been investigated [41]. In 2020, Behera et al. detected and classified drones in RGB images using the YOLOv3 network, and they achieved a mAP of 74% after 150 epochs. In this article, only drones were detected at various distances, and the issue of drone recognition and its distinction from birds was not discussed [42]. In 2020 Shi et al. proposed a detection process of the low-altitude drone based on the YOLOv4 deep learning network. They then compared the YOLOv4 detection result with the YOLOv3 and the SSD networks. In this study, the YOLOv4 network performed better than the YOLOv3 and the SSD networks in detecting, recognizing, and identifying three types of drones in terms of mAP and detection speed, achieving 89% mAP [43]. In 2021, Tan Wei Xun et al. detected and tracked a drone using the YOLOv3 deep learning network. In their study, the NVIDIA Jetson TX2 was used to detect drones in real-time. The results of this method show that the proposed YOLOv3 network detects drones of three sizes: small, medium, and large, with an average confidence score of 88% and a confidence score between 60% and 100% [44]. In 2021, Isaac-Medina et al. detected and tracked drones using a set of visible and thermal images and four deep learning network architectures. In this paper, the deep learning networks Faster RCNN, SSD, YOLOv3, and DETR (DEtection TRansformer) are used. Based on the results, all the studied networks were able to detect a small drone at a far distance. But the YOLOv3 deep learning network generally leads to better accuracy (up to 0.986 mAP) and the RCNN network performed better in detecting small drones (up to 0.77 mAP) [45]. In 2021 Singha et al. developed an automatic drone detection system using YOLOv4. They used a dataset of drones and birds to detect drones, and then evaluated the model on two types of drone videos. The results obtained in this study for detecting two types of multirotor drones are: mAP 74.36%, F1-score 0.79, recall 0.68, and precision 0.95 [46]. In 2021, Liu et al. examined three object detection methods, such as YOLOv3, YOLOv4, RetinaNet, and FCOS (Fully Convolutional One-stage Object Detector) networks, on visible image data. To get great accuracy in drone detection, the pruned YOLOv4 model is used to build a sparser, flatter network. The application of the method has improved the detection of small drones and high-speed drones. The pruned YOLOv4, with a pruning rate of 0.8 and a 24-layer pruning, achieved a mAP of 90.5%, an accuracy of 22.8%, a recall of 12.7%, and a processing speed of 60%. However, the challenges of crowded backgrounds, hidden areas, and surveys of multiple drone types have not yet been addressed [16]. In 2022, Samadzadegan et al. detected and recognized drones using YOLOv4 Deep Networks

in visible images [47]. This network can recognize multirotor and helicopters directly, and it can differentiate between drones and birds with a mAP of 84%, an IoU of 81%, and an accuracy of 83%. In this paper, the challenges related to recognition have been well addressed, but this method is limited to detecting and to recognizing only two drone types, such as multirotor and helicopter, and it has not detected other types [47].

In this study, to achieve higher accuracy in solving the challenges of drone type recognition in visible images, a modified YOLOv4 network is proposed. Drone recognition challenges include the drone's far distance from the camera, a crowded background, unpredictable movements, and the drone's resemblance to birds. As an independent approach, the proposed modified YOLOv4 deep learning network architecture is capable of recognizing birds and four types of drones: multirotors, fixed-wings, helicopters, and VTOLs. To show the improved results of the new model, its performance is also compared with the base YOLOv4 network.

## 4. Methodology

In this study, a modified network based on the latest version of the YOLO network is proposed. The steps to recognize bird species and four different types of drones are presented in Figure 4. In the first step, the input data was prepared to be ready to enter the proposed network. In the second step, the model was trained to recognize the drones, and the weight file obtained for the testing phase was generated. In the third step, the network was tested to observe how it worked; and, in the last step, the proposed deep learning network was evaluated using evaluation metrics.

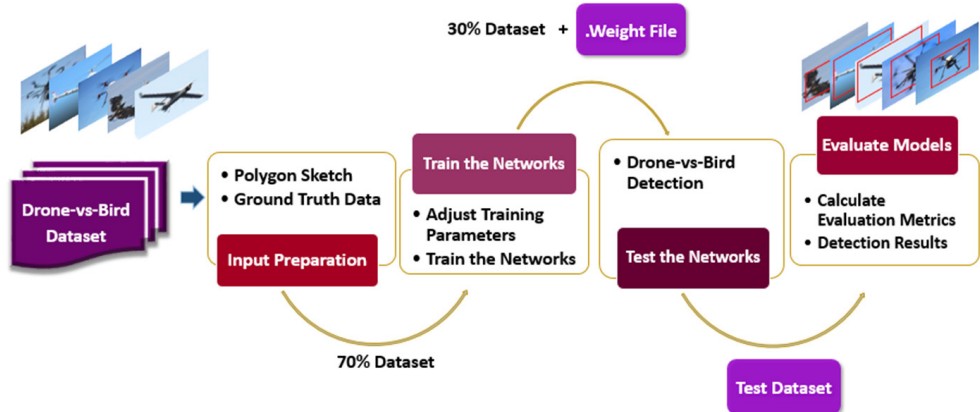

**Figure 4.** Recognition process using implemented modified network.

### 4.1. Input Preparation

Drones are generally divided into four categories: multirotors, fixed-wings, VTOLs, and helicopters. Multirotors are mainly developed in different structures such as octorotor, octo coax wide, hexarotor, and quadrotor. The fixed-wing drone can also be one of the types of fixed wing, plane a-tail, and standard plane, and the VTOL drones are the combination of both the previous versions, including four types of standard VTOL, VTOL duo tailsitter, VTOL quad tailsitter, and plane a-tail. Because of the similarity of the behavior of birds at long distances, this group of datasets belongs to the fifth category of network input data. The schematic drawings of each drone are presented in Figure 5. Thus, in this study, the data was labeled into a total of five classes and prepared for the training phase.

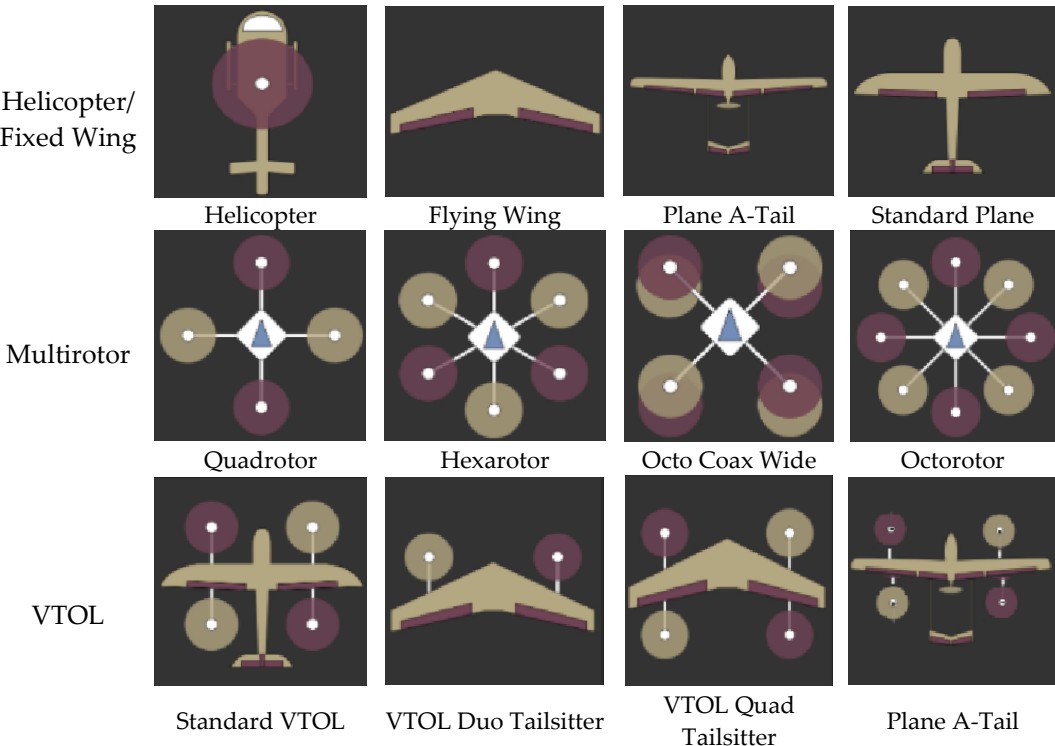

**Figure 5.** Different types of drones as Helicopters, Fixed-Wings, Multirotors, and VTOLs [48].

One strategy for preparing input data is to draw a rectangular bounding box around the object. In this study, the polygon sketch was used to label the drone dataset. Then, the best rectangle containing the object was fitted to the polygon. As shown in Figure 6, drawing the best rectangular bounding box around the drone results in the accurate extraction of the pixels containing the object.

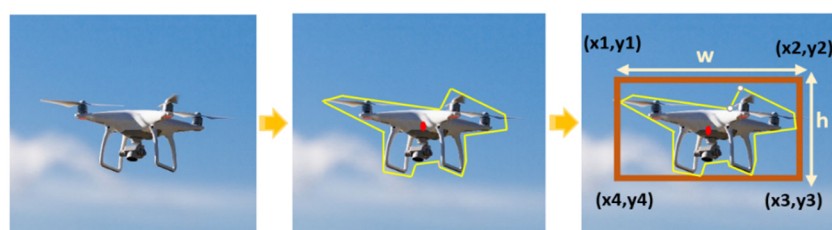

**Figure 6.** The conversion of the polygon to the best-fitted rectangle.

Finally, the bounding box central coordinates, the class number, and its width and its height are normalized in the range of [0, 1], and they are introduced to the next step in the proposed method.

### 4.2. Train the Networks

The YOLOv4 Deep Learning Network is selected as the drone vs. bird recognition network because of its advantages in this area. In addition, this network was modified to improve the performance of the basic network and to better address challenges.

#### 4.2.1. YOLOv4 Deep Learning Network Architecture

According to Figure 7, the YOLOv4 network consists of four main parts: the input of the network, backbone (feature map extractor), neck (feature map collector), and head (results of recognition).

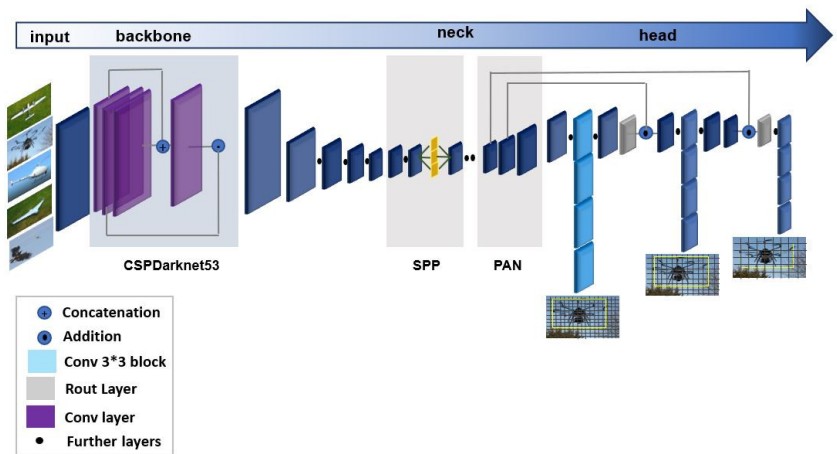

**Figure 7.** The basic YOLOv4 Deep Learning Network Architecture.

The backbone is responsible for extracting the feature map from the previously entered data [26]. In the implemented model, a Darknet53-based network, CSP Darkent-53 [49], is used to extract the feature. CSPDarknet53 enhances the learning process of CNN. The CSPDarknet53 overhead integration pyramid section is connected to improve the receiver field and to differentiate very important context features. Extracting better features leads to increasing the accuracy of detecting drones, recognizing their type, and differentiating them from birds. In the structure of the YOLOv4 network, there are several convolution layers after the backbone. In convolution layers, internal multiplication and feature extraction operations are performed using the obtained feature maps. In this network, a $3 \times 3$ convolutional layer is used after the backbone layer to extract more detailed and accurate features using the Mish activity function. The reason for using the Mish activity function is that this function also considers negative values, solving the problem of overfitting with precise regulatory effects.

The neck receives the feature map created in the backbone stage. This helps to add a layer between the backbone and the head. It consists of a modified Spatial Pyramid Pooling(SPP) and a modified Path Aggregation Network(PAN), both of which are used to gather information to improve accuracy [21,50]. Spatial Pyramid Pooling is an integration layer that removes the constraint of the fixed size of the network input. This layer consists of three pooling, with sizes 256-d, $4 \times 256$-d, $16 \times 256$-d, and (Figure 8). The SPP layer receives the feature map created from the previous convolution layer. It combines features and produces fixed-length outputs, it then connects to fully connected layers, and then enters the improved PAN network. This network is used to aggregate parameters for different detector surfaces instead of feature pyramid networks (FPNs) to detect the object used in YOLOv3.

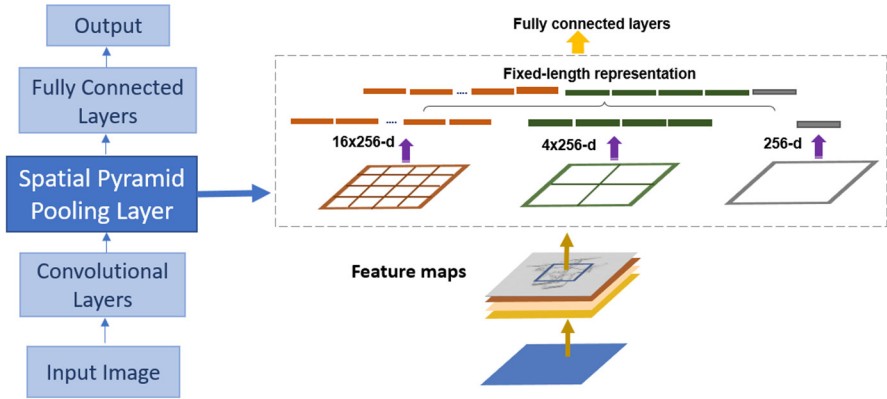

**Figure 8.** Spatial Pyramid Pooling (SPP) block.

The head part is used to classify and to locate the predicted boxes of the proposed network. At this level, the probabilities and the bounding box coordinates (x, y, height, and width) are given. This part uses the YOLOv3 deep learning network architecture, so the output is a tensor containing bounding box coordinates and class probabilities [51].

Bag-of-specials and bag-of-freebies are techniques used in the YOLOv4 algorithm to increase accuracy during and after training [26]. Bag-of-freebies helps to improve recognition during training, without increasing the inference time. This feature uses techniques such as data augmentation and drop blocks. Bag-of-freebies uses CutMin, data augmentation, and DropBlock techniques to increase and to regularize the data. It also uses techniques, such as grid sensitivity elimination, CIoU-loss, CmBN, self-adversarial training, and use of several anchors for one ground truth in detection techniques. Bag-of-specials are techniques that slightly increase the inference time and change the architecture. This feature uses methods such as non-maximum suppression, multi-input residual connections, and cross-stage partial connections (CSP) in the backbone. Additionally, in detection, it uses SPP-block, PAN-path aggregation, mish activation, SAM-block, and DIoU-NMS. Both technologies and their features are useful for training and testing the networks [26].

This network is not suitable for various applications, such as recognition of small objects and objects in cluttered backgrounds and hidden areas. Therefore, in this article, the YOLOv4 network architecture was modified to increase the accuracy of drone recognition and to improve the performance of the model to overcome challenges, which is one of the innovations in this article.

### 4.2.2. The Modified YOLOv4 Deep Learning Network Architecture

In the modified YOLOv4 network, according to Figure 9, three convolutional layers were added to the basic YOLOv4 network architecture after the backbone. Convolutional layers are useful for extracting features from images because these layers deal with spatial redundancy by weight sharing. The addition of these three layers makes the extracted features more exclusive and informative, and it reduces redundancy. This is primarily due to the repeated cascaded convolutions and to information compression by subsampling layers. By reducing redundancy, the network displays a compressed feature about the content of the image. Consequently, by increasing the depth of the network's convolutional layers, more accurate semantic features are extracted, recognition accuracy is increased, and overcoming existing challenges is facilitated [52,53].

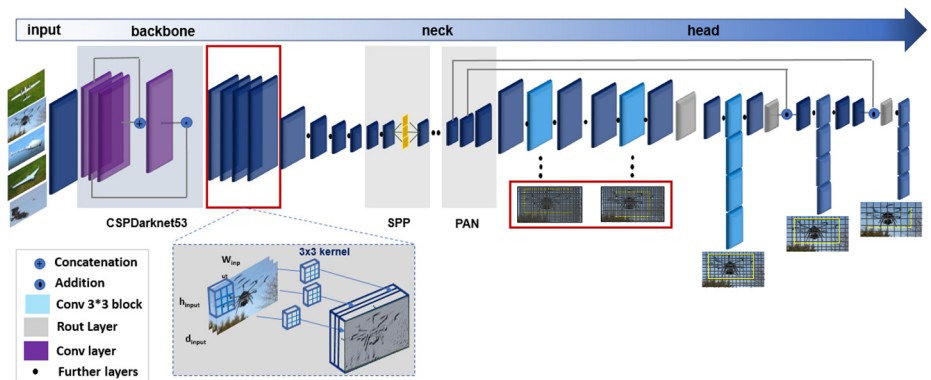

**Figure 9.** Recognition process using modified YOLOv4 network.

In addition, two large-scale convolutional layers were added to the head. In this part, a convolutional layer with a size of $3 \times 3$ and another with a size of $1 \times 1$ was used. This $1 \times 1$ convolutional layer reduced the final depth as well as the volume of network computation. This change balances the modified network to recognize large and small long-range targets.

The implemented network can recognize drones better than the basic YOLOv4 network at long distances with small sizes and against crowded backgrounds and hidden areas. This powerful object recognition model uses a single GPU to provide accurate object recognition. For training these two networks, 70% of the entire dataset was used. The two networks were trained under the same conditions, i.e., with the same data set, the same training parameters, and the same number of iterations. Once the model implementation and training phase are complete, the testing and evaluation phase of the network begins based on the selected evaluation metrics.

### 4.3. Test the Networks

After completing the network training, the network testing process began with 30% of the whole dataset. This stage was to select the best bounding box with the object and to evaluate the network's performance in recognizing drones and birds. The proposed learning network defines a bounding box around the detected drones. Since the drones in the dataset have different sizes and shapes, the proposed model creates multiple bounding boxes to recognize them. However, to select the most suitable bounding box out of the others, the non-maximum suppression (NMS) algorithm must be used [54]. This algorithm was used to remove bounding boxes with lower confidence scores and to select the best drone and bird box. As it appears from Figure 10, the green box is the best box that contains a drone, the other two bounding boxes also cover part of the drone, and they are candidates for drone recognition. In this algorithm, the bounding box with the highest confidence score is selected first and then the boxes with higher overlap with the selected box are removed. This process is continued until no representative boxes contain the drone. The same process is performed for the bird and, finally, the best bounding box is selected.

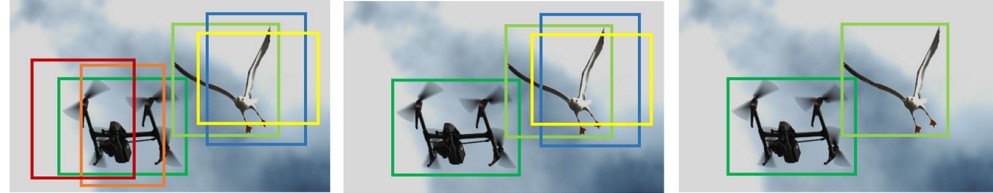

**Figure 10.** Proposed non-maximum suppression (NMS) box selection method.

The steps for performing the non-maximum suppression (NMS) algorithm can be summarized as follows:

- Select the predicted bounding box with the highest confidence level;
- Calculate *IoU* (the intersection and overlap of the selected box and other boxes) (Equation (1));

$$IoU = \frac{Area\ of\ Overlap}{Area\ of\ Union} \tag{1}$$

- Remove boxes with an overlap of more than the default *IoU* threshold of 7% with the selected box;
- Repeat steps 1–3.

In Figure 11, the above algorithm was run twice, and the green boxes were selected as the final bounding boxes containing the drone and the bird.

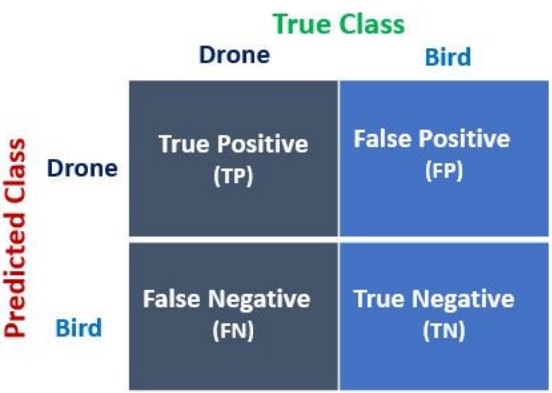

**Figure 11.** Schematic view of the confusion matrix.

*4.4. Evaluation Metrics*

To better understand how the proposed model works, the model was evaluated using mAP, IoU, accuracy, recall, F1-score, and precision.

The mAP is one of the most important evaluation metrics. In this paper, mAP was used as an evaluation metric to recognize drones and birds. It can be claimed that mAP is responsible for comparing the predicted bounding boxes by the network with the ground truth box [19].

The other important parameter that is the key to obtaining recall, F1-score, and precision measures is the confusion matrix. Computing confusion matrix can provide valuable information on the performance of the modified model and the presence of various errors [19]. Therefore, the values of recall, F1-score, and precision can be calculated from the confusion matrix using the values true negative (TN), false negative (FN), true positive (TP), and false positive (FP). The structure of the confusion matrix is illustrated in Figure 11.

The IoU shows the connection between the ground truth bounding box and the predicted bounding box. If the intersection value of these bounding boxes is above the default threshold value of 0.7, the classification is performed correctly (TP, true positive). On the other hand, if the IoU value is below 0.7, it is misdiagnosed (FP, false positive), and if these bounding boxes do not overlap with each other, it is considered a false negative (FN).

Precision means the percentage of positive predictions among predicted classes determined to be positive [19]. Precision, F1-score, and recall metrics are calculated separately for each class of multirotor, helicopter, fixed-wing, VTOL, and bird.

The recall value shows the percentage of positive predictions among all data in the positive class. The F1-score is the mean of values for accuracy and precision, and it can indicate the validity of the classification process. This metric works well for imbalanced data because it takes into account the FN and the FP values [19].

Another metric examined in this study is the overall accuracy of the model [19]. This metric shows the performance of the model in recognizing drones and birds.

## 5. Experiments and Results

To assess the performance of proposed networks in recognizing drones and distinguishing them from birds, the steps of implementing the YOLOv4 deep learning network and the proposed modified network are presented in this section. Moreover, the types of the applied dataset, the network evaluation over different types of images, and the recognition challenges are discussed.

*5.1. Data Preparation*

To begin the training phase, a set of 26,000 visible images were first prepared that included various bird species and four types of drones such as multirotors, helicopters, fixed-wing, and VTOLs (examples are shown in Figure 12). The use of these four drone types in the collected dataset is another innovation of this study. Public images and videos

were used to create the dataset. Approximately 70% of these images were used for training and 30% for testing the network. In both phases, the number of images used is the same in each class. To prepare these images, first, a polygon around the target was determined. This task was handled using an efficient and useful Computer Vision Annotation Tool (CVAT) and the input images were categorized into five classes. Based on these boxes, the best rectangles containing the drone were then selected and passed to the network. In this dataset, multirotors are classified in the first class, helicopters in the second class, fixed-wing aircraft in the third class, VTOL in the fourth class, and birds in the fifth class.

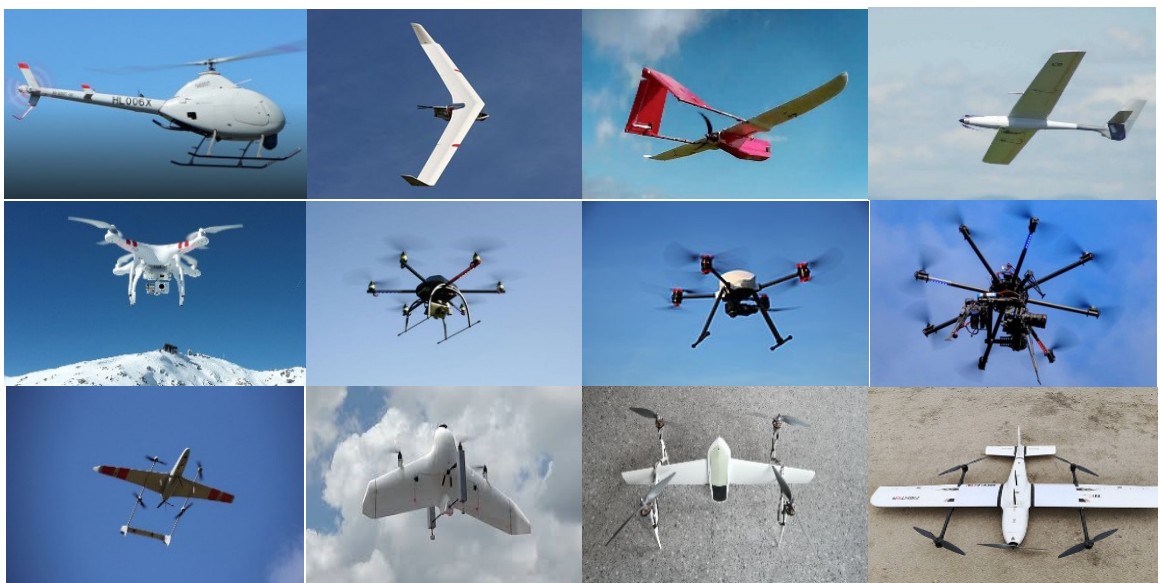

**Figure 12.** Included in the drone dataset.

### 5.2. Model Implementation and Training Results

The modified YOLOv4 network and the basic YOLOv4 network were trained using an Nvidia GeForce MX450 Graphics Processing Unit (GPU) hardware with 30,000 iterations. To train these two networks, the settings in the configuration file were changed as follows:

- The input image size was set to 160 × 160;
- The subdivision and the batch parameters were changed to 1 and 64, respectively (these settings were made to avoid errors due to lack of memory);
- The learning rate was changed to 0.0005;
- The step parameter was changed to 24,000 and 27,000, (with 80% and 90% of the number of iterations, respectively);
- The size of the filter in three convolutional layers near the YOLO layers was changed to 30 according to the number of classes.

The basic network used for training is the widely used Darknet Framework [55]. Depending on the hardware used, a CUDA (Compute Unified Device Architecture) Toolkit version 10.0, CUDNN (CUDA Deep Neural Network library) version 8.2, Visual Studio 2017, and OpenCV version 4.0.1 are used. During the training, a graph of the number of iterations and loss was plotted as shown in Figure 13. After 30 k iterations (about 3–4 days), the modified implemented method and the basic YOLOv4 model achieved a loss of 0.58 and 0.68, respectively. The obtained values show that the detection loss of the proposed network was lower than that of the basic network, indicating the better performance of the proposed model in the training stage.

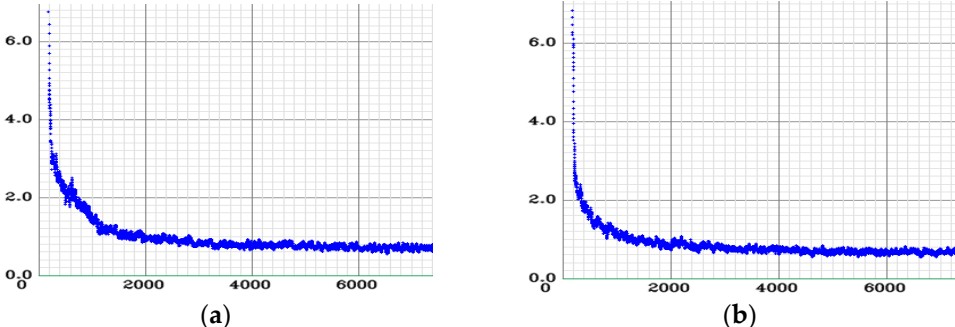

**Figure 13.** The loss graph in the training process; (**a**) The YOLOv4 network. (**b**)The modified YOLOv4 network.

### 5.3. Evaluation of the YOLOv4 and Modified YOLOv4 Models

The modified YOLOv4 and the basic YOLOv4 networks were precisely evaluated using accuracy, mAP, recall, precision, and F1-score in recognizing the drones. Tables 1 and 2 show the evaluation result of the proposed model and the basic YOLOv4 network. Accordingly, the performance of both models was compared with the same dataset, with an IoU default threshold of 0.07 and the same number of iterations. The recognition performance of the modified model was performed in five classes: multi-rotor, helicopter, fixed-wing, VTOL, and birds; and, the evaluation metrics of precision, recall, F1-score, accuracy, mAP, and IoU were increased. These results indicate the better performance of the modified model than the basic YOLOv4 model. In this network, the accuracy was 0.83%, the mAP was 83%, and the IoU was 84%, which was 4% better than the basic model.

**Table 1.** Evaluation results of the basic and Modified YOLOv4 networks.

| Dataset | Model | Num of Images | Precision % | Recall % | F1-Score % |
|---|---|---|---|---|---|
| Bird | YOLOv4 | 1570 | 81 | 87 | 84 |
| | Modified YOLOv4 | | 87 | 90 | 89 |
| Fixed Wing | YOLOv4 | 1570 | 88 | 70 | 78 |
| | Modified YOLOv4 | | 88 | 77 | 82 |
| Helicopter | YOLOv4 | 1570 | 81 | 73 | 77 |
| | Modified YOLOv4 | | 88 | 73 | 80 |
| Multirotor | YOLOv4 | 1570 | 77 | 90 | 83 |
| | Modified YOLOv4 | | 79 | 90 | 84 |
| VTOL | YOLOv4 | 1570 | 72 | 77 | 74 |
| | Modified YOLOv4 | | 74 | 83 | 78 |
| Total | YOLOv4 | 7850 | 80 | 79 | 79 |
| | Modified YOLOv4 | | 83 | 83 | 83 |

**Table 2.** Total evaluation results of the basic and Modified YOLOv4 networks.

| Dataset | Model | Num of Images | Accuracy % | mAP % | IoU % |
|---|---|---|---|---|---|
| Total | YOLOv4 | 7850 | 79 | 79 | 80 |
| | Modified YOLOv4 | | 83 | 83 | 84 |

The comparison of precision, recall, F1-score, and total mAP in all five classes in the two implemented models are graphically presented in Figure 14. The precision was improved in four classes (multirotor, helicopter, VTOL, and bird) and unchanged in the fixed-wing class. In general, these results show the improvement of model performance in

the proposed network. By comparing the recall according to Figure 15, its improvement is observed in three classes, and there is no change in the other two classes. This means that the ability and the performance of the second model in recognition were improved. As it appears from Figure 16, the F1-score increased compared to the basic model of YOLOv4, and this also shows the better performance of the modified YOLOv4 network.

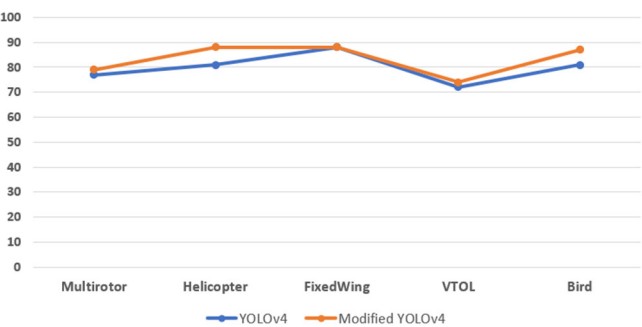

**Figure 14.** The comparison between the precision values in the basic and the modified YOLOv4 models.

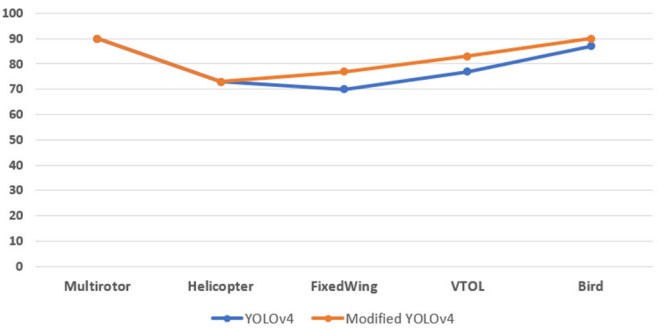

**Figure 15.** The comparison between the recall values in the basic and the modified YOLOv4 models.

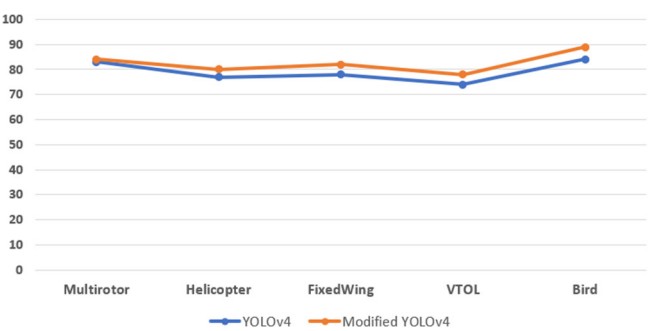

**Figure 16.** The comparison between F1-score in the basic and the modified YOLOv4 models.

Figure 17 shows the results of drone recognition in the basic YOLOv4 network and Figure 18 shows these results with the modified YOLOv4 network. A comparison of the results of the two networks shows an improvement in the accuracy of the bounding boxes and the class probabilities in the modified network. On the other hand, the recognition of four types of multirotor (octorotor, octo coax wide, hexarotor, and quadrotor), three types of fixed-wing (flying wing, plane a-tail, and standard plane), four types of VTOLs (standard VTOL, VTOL duo tailsitter, VTOL quad tailsitter, and plane a-tail), one type of helicopter, and their discrimination from birds were improved in the proposed model.

| Dataset | Sample 1 | Sample 2 |
| --- | --- | --- |

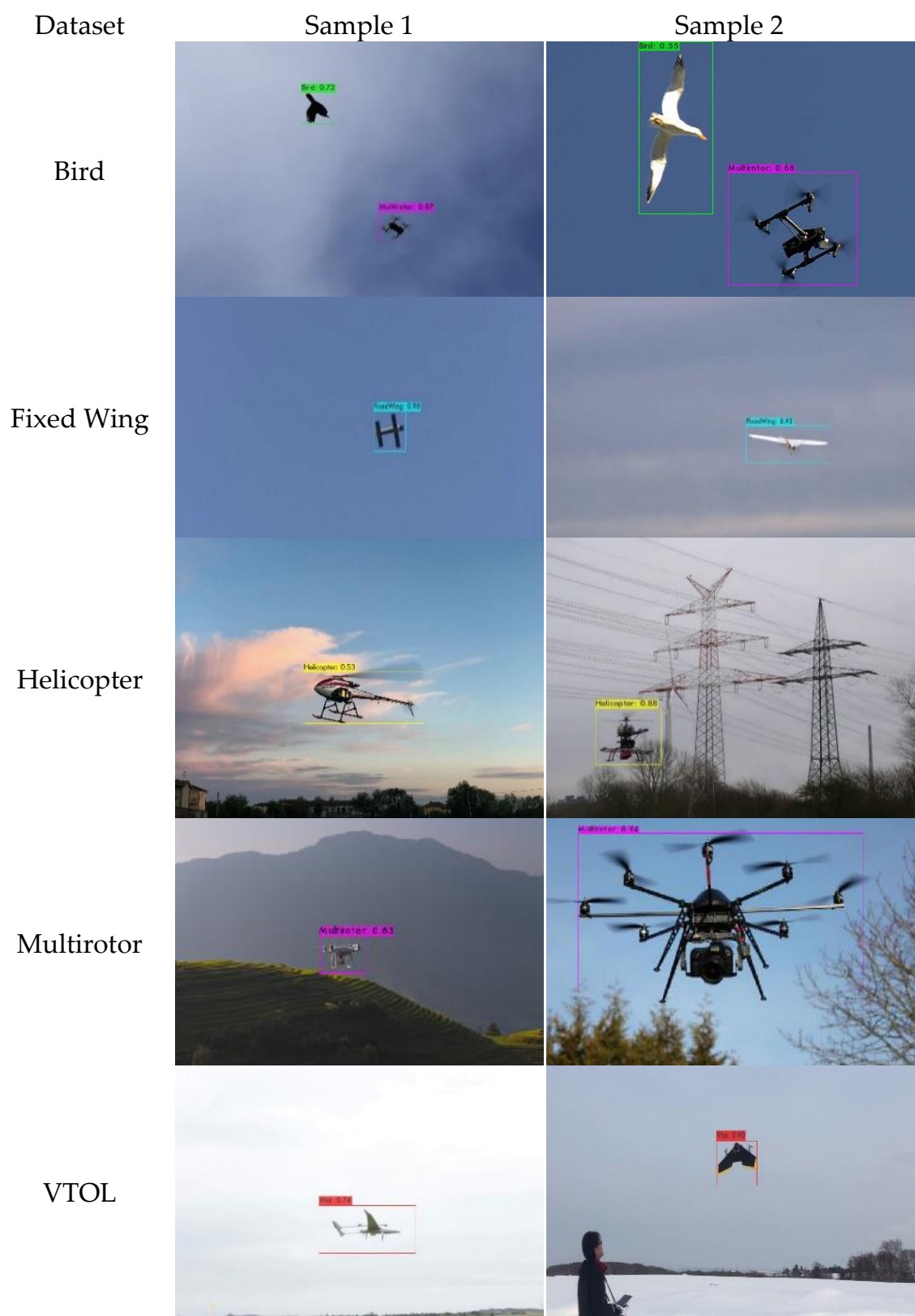

**Figure 17.** Some samples of drone-vs-bird recognition results with a basic YOLOv4 network.

| Dataset | Sample 1 | Sample 2 |

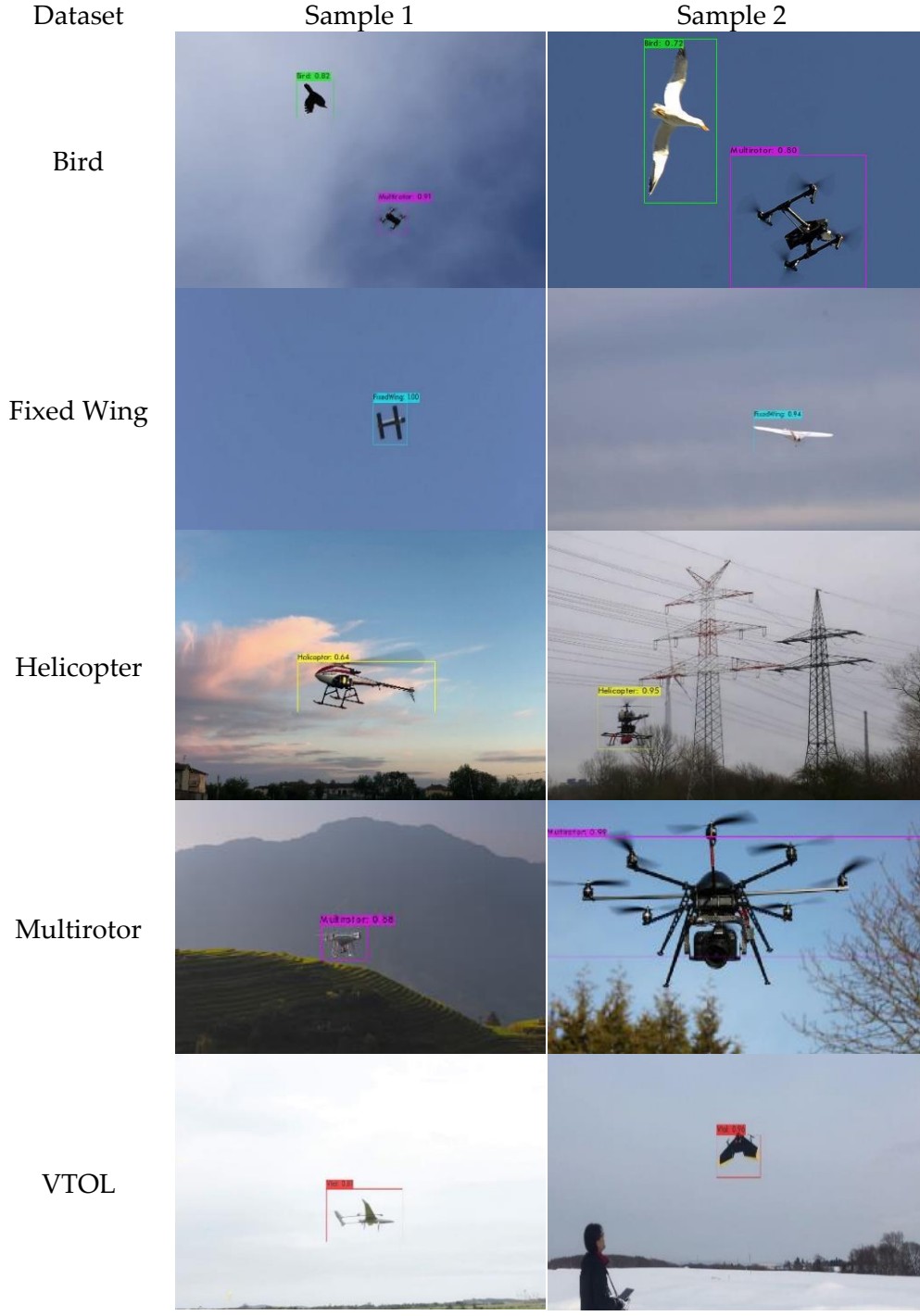

**Figure 18.** Some samples of drone-vs-bird recognition results with a modified YOLOv4 network.

*5.4. Addressing the Challenges in the Modified YOLOv4 Model*

In this study, the YOLOv4 network was modified to improve the recognition results of four types of drones and to differentiate them from birds concerning existing challenges, such as the smaller size of the drones, crowded backgrounds, loss of scalability, and similarity with birds. Figure 19 shows the performance of both the basic and the modified models in overcoming the challenges.

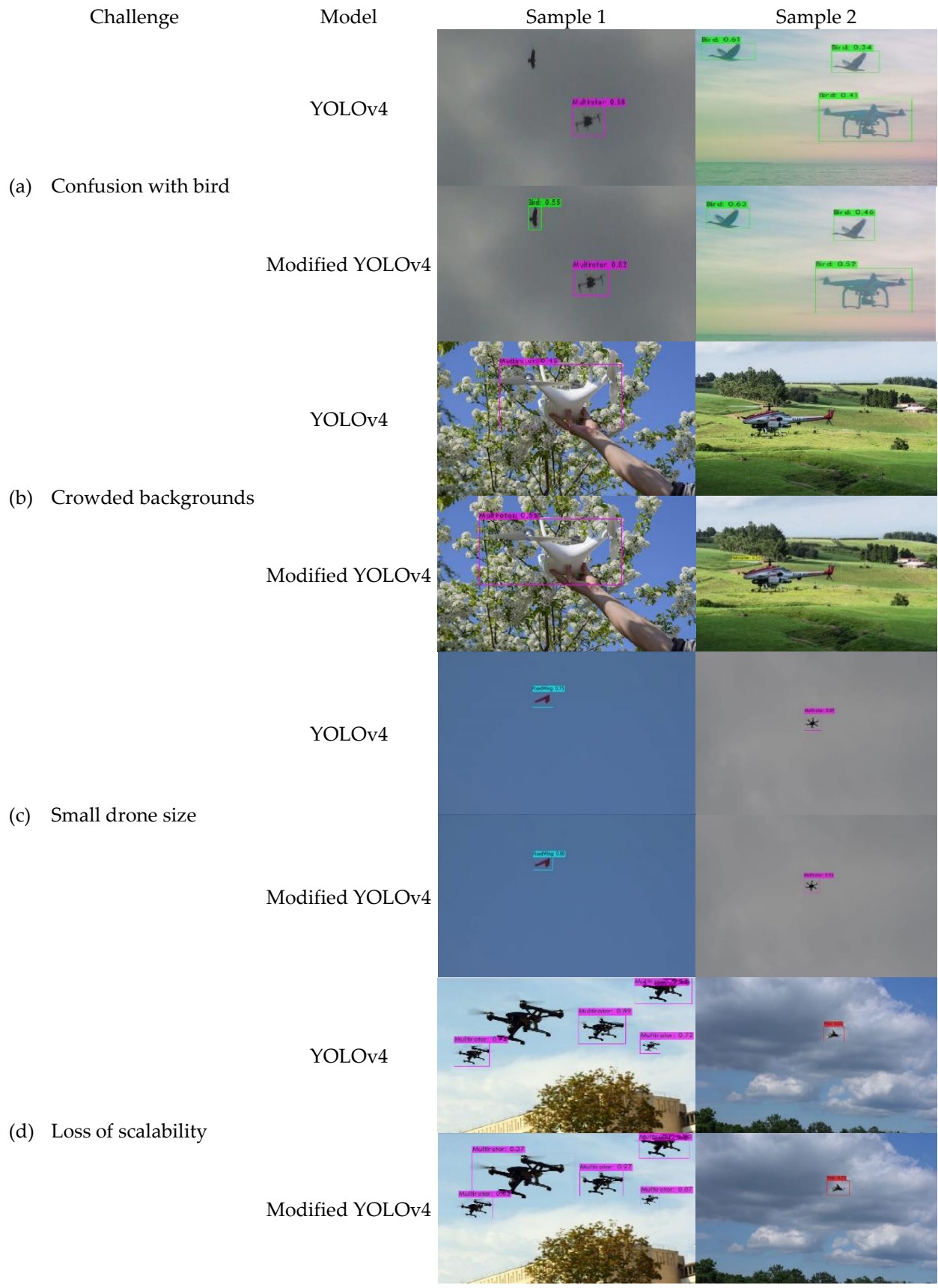

**Figure 19.** Samples of handling challenges in drone recognition.

The first challenge is to recognize the drone and to distinguish it from the bird; the basic model is unable to recognize the bird in some samples and the class probability values are lower than for the modified model. The modified YOLOv4 model can recognize drones and distinguish them from birds by extracting smaller and more accurate features, and it has demonstrated this ability in the evaluated samples (Figure 19). The second challenge studied is the presence of drones in a crowded background, where the modified model can recognize targets with a high prediction probability. In some cases, the base model is unable to recognize the drone or it has a lower probability than the modified model. The third challenge is the small size of the drones and their placement at long ranges, where the modified model performs better than the base model. The modified model can accurately recognize all small drones in the test images. This ability is due to the use of more convolutional layers at the head of the network. The change in network structure does not remarkably affect the execution time of the network training algorithm. The training of the proposed network takes approximately 3 h longer than that of the first network. Finally, the challenge of the detectability swarm of UAVs at different scales is investigated. In this challenge, the base model is unable to recognize drones in the images in some cases, and it generally has lower class probability than the modified model.

Figure 20 illustrates several samples of different drone types in crowded environments with lighting conditions and different weather. As the results show, the modified YOLOv4 network is capable of recognizing different drone types under complex and challenging conditions.

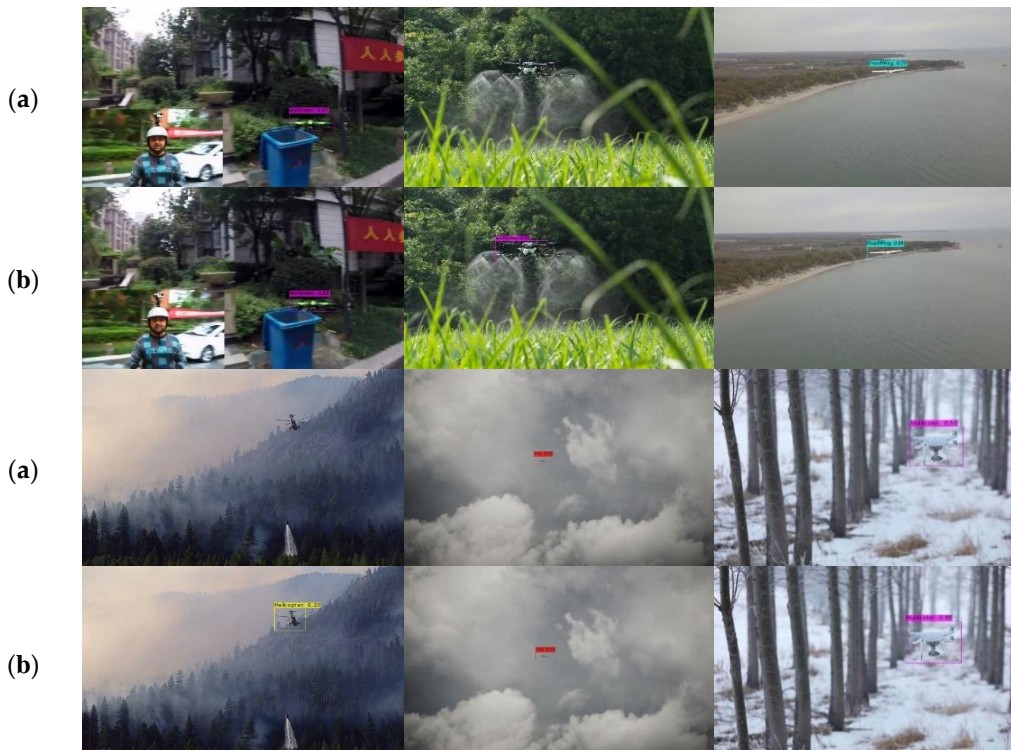

**Figure 20.** Several samples of drone recognition in different light and weather conditions and against a crowded background. (**a**) YOLOv4. (**b**) Modified YOLOv4.

## 6. Discussion

The evaluation metric examined in this study is confusion matrix, IoU, mAP, accuracy, precision, recall, and F1-score. The values obtained from these criteria in the modified network are as follows: 79% precision, 84% F1-score, and 90% recall for the multirotor; 88% precision, 82% F1-score, and 77% recall for the fixed-wing; 88% precision, 80% F1-score, and 73% recall for the helicopter; 74% precision, 78% F1-score, and 83% recall

for VTOL; and 87% precision, 89% F1-score, and 90% recall for the bird. Based on the basic YOLOv4 network results, it can be said that the proposed network had an average improvement of 3.4% in precision, 3.2% in the recall, and 3.4% in F1-score values. The overall accuracy, indicating the correct classification of the input data set into five classes, was also investigated. This criterion increased by 4% in the modified network compared to the basic network. The mAP also increased by 4% in the modified network compared to the basic YOLOv4 network. This rate indicates an improvement in the classification of the proposed model in the five classes. Finally, the IoU metric was used to calculate the overlap value of the predicted bounding box with the ground truth bounding box. This metric also increased by 4% in the modified network, indicating the higher accuracy of the predicted bounding box.

In recent years, the use of artificial intelligence and deep learning methods has become one of the most popular and useful methods in object recognition. In 2021, Tan Wei Xun et al. detected and tracked a drone using the YOLOv3. The results of this method show that the proposed YOLOv3 network detects drones of different sizes with a confidence level between 60 and 100% [44]; also this year, drone detection and tracking were performed by Isaac-Medina et al. using a set of visible and thermal images and four deep learning network architectures, including RCNN, SSD, YOLOv3, and DETR. According to the results of this study, it can be said that the YOLOv3 deep learning network performs better than the other models [45]. One of the problems with these studies was the inability to detect small drones over long distances. In addition, an automatic drone detection system using YOLOv4 was developed by Singha et al. In this system, a collection of drone-vs-bird images were used to detect the drone and its evaluation was tested using a video dataset. The results of this work include the high accuracy of the YOLOv4 network [46]. One of the limitations of this study was the lack of recognition of the different types of drones and the lack of challenging images. Moreover, Liu et al. applied three object detection methods, such as YOLOv3, YOLOv4, RetinaNet, and FCOS networks on the drone dataset. To build a scattered, flat network and to get great accuracy in drone detection, the pruned YOLOv4 model was used, which improved the accuracy of drone detection at high speed. However, the study did not address the challenges of crowded background images, hidden areas, and the distance of the drone from the camera [16]. Additionally in 2022, Samadzadegan et al. recognized two types of drones (multirotors and helicopters) and birds using the YOLOv4 deep learning network. However, this paper did not recognize two other drone types such as VTOL and fixed-wing drones, which could be more dangerous than the previous two types. In addition, a rectangular bounding box was used in the data preparation, which resulted in the input of additional information and reduced the accuracy of recognition [47].

Previous studies have focused on drone detection, and a small number of these studies have examined the recognition of different types of drones. Moreover, the challenges of drone recognition, such as the small size of the drone, crowded background, hidden areas, and confusion with birds, have not been comprehensively addressed in these studies. Therefore, it can be said that the unauthorized presence of drones in challenging environments and their inaccurate recognition in sensitive infrastructures is still one of the most important problems in ensuring public safety. The main goal of this research is to recognize four types of drones and to differentiate them from birds at far distances despite challenges, such as the small size of the drone, a crowded background, and the presence of hidden areas.

In this study, the YOLOv4 network was modified to improve drone recognition challenges. A set of visible images with different types of drones and birds in different environments at near and far distances were collected to recognize four types of drones and to differentiate them from birds. Two convolutional layers were added to the head of the YOLOv4 to solve the challenges of small drone recognition. For example, in Figure 20 there are examples of small drones that the basic model was unable to recognize in some cases. However, the modified model recognized them; and, in other cases they operated

with less accuracy than the modified network. This result shows the improvement of the current network compared to the basic network for recognizing small drones. Furthermore, to increase the accuracy of feature extraction and precise recognition of drones and birds, three convolutional layers were added after the backbone layer. Adding these layers to the architecture of the basic YOLOv4 network did not change the training time of the network, and it took only a few hours longer than training the basic network. By applying these changes in the network architecture and using extensive datasets, the proposed method was able to recognize all drone types and bird species in challenging environments. In Figure 19, rows (a), (b), and (d) provide examples of difficult images and swarms of UAVs, all of which have higher recognition accuracy in the modified network than in the basic network. Figure 20 also contains other challenging examples where the proposed network performs better.

### 7. Conclusions

As has been noted, UAV recognition in various situations is a complex process; the usual methods and even conventional deep learning network methods do not work well in some cases. In this study, the basic YOLOv4 network was used to recognize drone types and to differentiate them from bird species. To increase recognition accuracy and to better address existing challenges, a novel and modified model of this network was proposed. To train, test, and evaluate these two networks, a collection of 26,000 visible image datasets including four types of UAVs (multirotor, fixed-wing, helicopter, and VTOL) and birds were collected. The comparison of these two models was done using mAP, confusion matrix, IoU, precision, accuracy, F1-score, and recall evaluation metrics. With the modified YOLOv4 model, we achieved 84% IoU, 83% mAPs, and 83% accuracy, which is better than the basic model, and it solved the challenges well. In the future, real-time identification with onboard systems can be studied in addition to drone recognition. Background removal algorithms can also be used to make labeling input data easier and faster. In addition to the multirotor types used, Tri-rotors can also be used to complete the dataset. Additionally, other deep learning networks can be used to compare their results with the results of this modified network.

**Author Contributions:** All authors contributed to the study conception and design. F.D.J. contributed to supervision, reviewing, and validation. F.S. is a drone expert involved in the conceptualization, methodology, and editing of the draft. F.A.M. contributed to programing, visualization, computer vision concepts, writing, and editing. M.G. is involved in software, deep learning concepts, data collection and preparation, and drafting. All authors have read and agreed to the published version of the manuscript.

**Funding:** This research received no external funding.

**Institutional Review Board Statement:** Not applicable.

**Informed Consent Statement:** Not applicable.

**Data Availability Statement:** Not applicable.

**Conflicts of Interest:** The authors have no conflict of interest to disclose.

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
