# Peer review of "A Modified YOLOv4 Deep Learning Network for Vision-Based UAV Recognition"

_drones, doi:10.3390/drones6070160_

Round 1

Reviewer 1 Report

The paper is well written and structured and the authors well followed the advice given therefore thewell followed each advice. According to the present reviewer the manuscript can be published on MDPI Drone on the present form.

Author Response

Thank you very much for all the constructive comments and support.

Reviewer 2 Report

This paper addressed the problem of detecting UAVs by employing CNNS. The authors modified YOLOv4 network by adding a few layers. They demonstrated an improvement in accuracy by employing their proposed network. They modified the way that the patches are selected, by considering better regions that have a lower number of false cases. There are many typos, erros, and statements that need to be revised in the article, so I recommend the journal to accept the article after the authors do a major revision. Here are some of my comments.

Line 15: You used  a contrast which is not correct in this context.

Line 16: Why didn't you include the airport and power plants as important buildings? This listing seems odd to me.

Line 18: Confusion with birds is a challenge? How? We can say that unauthorized flying of UAVs is a challenge.

Line 22: “Recently, the use of …. is very important” Very bad English statement.

Line 25: “This network”: Which network?

Line 34: “A slight” why did you use capital letters?

Why don't you present F1-Score as a result? What are the bases for the accuracies and precisions? Training test validation?

The structure of the introduction needs to be improved. It seems to be very naïve and general, but then it attacks some specific problem. The structure is now similar to the Introduction, not an abstract.

Line 50-53: The terms detection, recognition and and identification used in an old-fashioned and peculiar way in my opinion. Detection and recognition used to detect a flying object from other static background objects and recognizing the flying object to be a drone, and finally identifying its subtype. All those separations are usually united in a CNN.

Line 54: Why don't you identify the sub-type either? From the abstract I understood that the subclass is also detected.

Line 56: All abbreviations should be expanded upon their initial use: radar, Lidar, etc.

Line 42-67: very long paragraph. I advise you to break in and organize it in a better way.

Line 62: “detection and recognition of purposes” What does this combination mean? I advise you to read the manuscript several times to find many faults that you made.

Line 61-63: The reason is not justified by the way that the author selected. You need to clarify your purpose in an easy to understand way.

Line 74: Also image locations of drones are recognized by deep learning methods.

Line 77: This is a very vague statement. I recommend you to modify it.

Line 88: It seems that you have two steps here, one is applying convolutional filters and then feature extraction. Could you please tell which operation you intend for feature extraction?

Line 88: “performed higher accuracy” please correct.

Line 96: why do you expand CNN here?

Line 100: In my opinion, this is a very strong claim without sufficient evidence. You must either bring evidence or add suitable citations.

Line 100-103: The challenges that you mentioned could be addressed by a better training set. I expected to see the novelty of your work up to this point. It seems that you want to revolutionize the way CNN works, but in contrast I don’t see any significant improvement in the network that you have proposed. I recommend that you state logical claims that will be judged based on the presentation that you have prepared.

Line 106: What is VTOL? You should expand all the abbreviations upon their first usage.

Line 112: No need to repeat.

Line 133: smaller than what???? The comparison is not complete.

Line 140: Very bad sentence. What do you mean by “different resolution”? Are you talking about cameras or drones? What do you mean by different scales? It is very obvious that drones come with different sizes and shapes. The reasons to highlight the difficulty of this problem could be zipped into one paragraph in my opinion. The authors are over-stating it.

Line 155-159: Many citations are missing.

Line 161: What is mAP? You should better discuss the presented networks. When you bring classification metrics or some performance metric, they should be the same between the comparing list, so the reader could possibly judge them. The way that you chose is confusing and non informative.

Line 224: “These challenges” which challenges?

Line 234: You should first describe your proposed network, and then discuss about its benefits. It is confuzing that you are stating the benefits of your proposed approach before demonstrating it to the readers.

Line 261: How do you draw a bounding box in a shape? Very inaccurate statement.

Line 262-271: I recommend you to rewrite this part.

Line 263: How “drawing” can increase “the number of unrelated pixels”???? Very bad statement.

Line 347-358: Why do you add 3 convolution layers? Will adding more layers help? The modification over the original network is minimal. There should be justification for this modification. It is understandable that adding more parameters will help the network to act better, but what is the impact of this modification on the other factors such as performance?

Line 368: I don’t see any reason why your modification makes the original CNN optimized for parallel computing. The original CNN was already suitable for parallel computing. This is an over statement and inaccurate description of your modification.

Line 374: here you jumped to after training. Why is the training specification missed? How many percent of the dataset was used for training, and how much for tests? Was the training fair for both networks? Was there a validation set?

Line 399: What is IOU?

Line 409: Why don't you expand mAP on its first usage?

Line 520: Why are some cells not filled?

Author Response

Thank you very much for reviewing our manuscript. We also greatly appreciate the reviewers for their complimentary comments and suggestions. We have carried out the experiments that the reviewers suggested and revised the manuscript accordingly. We hope that you find the responses satisfactory and that the manuscript is now acceptable for publication.

Round 2

Reviewer 2 Report

Despite the fact that the novelty is still questioning, I see that the authors made a small improvement over the original network. The edits are satisfactory to me, therefore, I recommend the journal to accept this article in the currently modified format. 

This manuscript is a resubmission of an earlier submission. The following is a list of the peer review reports and author responses from that submission.

Round 1

Reviewer 1 Report

The paper is well written and well structured. The title can be improved to catch up more the attention according the present reviewer such as: 

A new tentative modified YOLOv4 Deep Learning Network for Vision-Based UAV Detection and Recognition. This can better remark the novelty proposed into the present manuscript.

I the template is not indicated if it is an article or other I suppose an article please correct this. Also the images are not formatted according MDPI Drones.

I suggest also to introduce in the introduction section some information about application of similar approaches in a combined use with satellite and drones. I suggest to consider these works:

1) https://doi.org/10.3390/rs12213542

2) https://doi.org/10.12895/jaeid.20172.690 

Finally the confusion matrix is not well clear and it doesn't provide crucial information like overall accuracy, commission and omission errors I suggest you to consider to introduce them in a table.

Author Response

Response to reviewers' comments

 The Drones journal

Ms. Rocksy Zhang:

Thank you very much for reviewing our manuscript. We also greatly appreciate the reviewers for their complimentary comments and suggestions. We have carried out the experiments that the reviewers suggested and revised the manuscript accordingly. We hope that you find the responses satisfactory and that the manuscript is now acceptable for publication.

Reviewer 1

We appreciate that the reviewer’s comments. In the followings are our point-by-point responses:

  1. The title can be improved to catch up more the attention according the present reviewer such as: A new tentative modified YOLOv4 Deep Learning Network for Vision-Based UAV Detection and Recognition. This can better remark the novelty proposed into the present manuscript.

Response: Thank you for your constructive comments. In response to your suggestion, we changed the title of the article to your suggestion.

-------------------------------------------------------------------------------------------------------------------------------

  1. I the template is not indicated if it is an article or other I suppose an article please correct this. Also the images are not formatted according MDPI Drones.

Response: According to the comment, these points are applied in all images and tables.

-------------------------------------------------------------------------------------------------------------------------------

  1. I suggest also to introduce in the introduction section some information about application of similar approaches in a combined use with satellite and drones. I suggest to consider these works:

1) https://doi.org/10.3390/rs12213542

2) https://doi.org/10.12895/jaeid.20172.690 

Response: Thank you for introducing these articles. We have reviewed and referenced these articles in the introduction section on lines 43 to 45.

-------------------------------------------------------------------------------------------------------------------------------

  1. Finally the confusion matrix is not well clear and it doesn't provide crucial information like overall accuracy, commission, and omission errors I suggest you to consider to introduce them in a table.

Response: The confusion matrix itself does not provide important information to the reader. Rather, by analyzing and calculating it, the evaluation metrics of mAP, IoU, accuracy, recall, F1-score, and precision are obtained. For this reason, we have removed the confusion matrix and displayed the final results in Table 1 instead.

-------------------------------------------------------------------------------------------------------------------------------

Reviewer 2 Report

The authors focus on a relevant problem, which is still an open one.

The structure of the paper is a common one but it is not well concretised. 

The research uses a state of the art neural network to perform detection and suggests a modified version of this network to achieve better performance. I think that the authors are very candid about the modifications that are suggested. From my understanding, the difference from the improved network to the standard yolo v4 is the addition of 3 conv layers in the backbone and 2 conv layers in the head part of the network. These modifications are mentioned along with the description of standard elements of the yolo v4 network and which makes it harder for the reader to have a clear understanding of the contribution of the paper.

I believe that adding more convolutional layers is a relatively common practice to marginally improve results, so I guess the contributions are only incremental. Nonetheless, if the authors believe that this is important, then it should be presented in a clearer way, highlighting these modifications and not disguising them in the general description of yolo v4.

The description of the overall pipeline which is done in fig 5 is also confusing. Several of  the blocks have redundant items such as: 

-“training the Yolo v4 and the modified network” and “train the networks”

-”Model evaluation” and “evaluate models”

There are also some items which are not clear at all (e.g. “prepare networks”)

Fig. 6 is also important but should be improved, since it depicts examples of UAVs belonging to the classes to be detected. The examples should be grouped into these classes.

The final part of section 3 also describes in great detail several performance parameters which are pretty used in the community. I believe that this description was not completely necessary, since they are commonly used. Additionally, I also believe that too many performance parameters are considered. It is quite common to use Average precision to encompass the overall performance.

The authors also discourse on the labeling procedure, arguing that labeling a polygon around the object of interest allows to create a better enclosing bounding box than creating a bounding box directly. I believe that this is not obvious at all and highly discussable.

Section 4 should also be improved.

Figures 18, 19 and 20 are supposed to highlight the performance of the modified network  over the standard one. Unfortunately, the images are too small in the manuscript and the labels that they contain are completely unreadable.

The manuscript has too many typos and style problems. This is demonstrative of a lack of care in the preparation and review. I will list a few just to  highlight the magnitude of the problem:

-Pag6 line 235 “... Plane A-.”

-Citations should have a space separating them from the preceding word. Example Pag. 2 lines 53 to 55.

-Pag. 9 line 316  “....and the head and It consists….”

Author Response

Response to reviewers' comments

 The Drones journal

Ms. Rocksy Zhang:

Thank you very much for reviewing our manuscript. We also greatly appreciate the reviewers for their complimentary comments and suggestions. We have carried out the experiments that the reviewers suggested and revised the manuscript accordingly. We hope that you find the responses satisfactory and that the manuscript is now acceptable for publication.

Reviewer 3

  1. The research uses a state of the art neural network to perform detection and suggests a modified version of this network to achieve better performance. I think that the authors are very candid about the modifications that are suggested. From my understanding, the difference from the improved network to the standard yolo v4 is the addition of 3 conv layers in the backbone and 2 conv layers in the head part of the network. These modifications are mentioned along with the description of standard elements of the yolo v4 network and which makes it harder for the reader to have a clear understanding of the contribution of the paper.

Response 1: Thanks for your constructive comment. We have added these modifications in a separate section called "The Modified YOLOv4 Deep Learning Network Architecture" in lines

 346 to 367.

--------------------------------------------------------------------------------------------------------------------------

  1. I believe that adding more convolutional layers is a relatively common practice to marginally improve results, so I guess the contributions are only incremental. Nonetheless, if the authors believe that this is important, then it should be presented in a clearer way, highlighting these modifications and not disguising them in the general description of yolo v4.

Response: Improving the network by adding the layers mentioned in this study has resulted in higher accuracy in UAV detection and recognition. As presented in Figure 19, row (C), sample 2, the YOLOv4 baseline network and the modified network are able to recognize the drone with 89% and 96% accuracy, respectively. Also, Figure 20 shows more challenging examples where the performance of the modified network is improved compared to the basic network. According to these examples and the results of the network modifications, these changes, and additional layers are acceptable for our study.

We added these points in lines 661 to 666 and 673 to 677.

-------------------------------------------------------------------------------------------------------------------------------

  1. The description of the overall pipeline which is done in fig 5 is also confusing. Several of the blocks have redundant items. 

“training the Yolo v4 and the modified network” and “train the networks”

-”Model evaluation” and “evaluate models”

There are also some items which are not clear at all (e.g. “prepare networks”)

Response: These points have been corrected in fig 5.

-------------------------------------------------------------------------------------------------------------------------------

  1. 6 is also important but should be improved, since it depicts examples of UAVs belonging to the classes to be detected. The examples should be grouped into these classes.

Response: According to this comment, Figure 6 in the modified version of the article has been modified. To better understand the Figure, we added a column to specify the type of drones.

-------------------------------------------------------------------------------------------------------------------------------

  1. The final part of section 3 also describes in great detail several performance parameters which are pretty used in the community. I believe that this description was not completely necessary, since they are commonly used. Additionally, I also believe that too many performance parameters are considered. It is quite common to use Average precision to encompass the overall performance.

Response: Thank you for this comment. We have removed the formulas related to the evaluation metrics and reduced the description. We have also removed the confusion matrix criterion. This is because the analysis of the results is fully explored and its existence does not bring useful information to the reader.

-------------------------------------------------------------------------------------------------------------------------------

  1. The authors also discourse on the labeling procedure, arguing that labeling a polygon around the object of interest allows to create a better enclosing bounding box than creating a bounding box directly. I believe that this is not obvious at all and highly discussable.

Response: The reason for the superiority of this method comparing to drawing a rectangle is explained in lines 271 to 275.

-------------------------------------------------------------------------------------------------------------------------------

  1. Section 4 should also be improved.

Figures 18, 19 and 20 are supposed to highlight the performance of the modified network  over the standard one. Unfortunately, the images are too small in the manuscript and the labels that they contain are completely unreadable.

Response: According to this comment, we have removed the poor-quality images from the figures and enlarged the dimensions of the images to provide more clear view.

-------------------------------------------------------------------------------------------------------------------------------

  1. -Pag. 9 line 316  “....and the head and It consists….”

Response: This point has been corrected.

-------------------------------------------------------------------------------------------------------------------------------

  1. -Citations should have a space separating them from the preceding word. Example Pag. 2 lines 53 to 55.

Response: This point is corrected throughout the article in similar cases.

-------------------------------------------------------------------------------------------------------------------------------

  1. -Pag6 line 235 “... Plane A-.”

Response: This point is corrected in line 253.

-------------------------------------------------------------------------------------------------------------------------------

  • In addition to the above corrections, the text has been revised and many style problems and vague sentences have been corrected.

-------------------------------------------------------------------------------------------------------------------------------

-------------------------------------------------------------------------------------------------------------------------------

Reviewer 3 Report

This article proposed a new approach to detect UAVs from background and recognize the sub classes. The topic is very interesting. Unfortunately, the article suffers from many different problems. More importantly, the scientific novelty is not clear to me. The authors claim that they have modified Yolo V4, but the modification details were not appropriately discussed. There is a need, in my opinion, for an intensive English review and fixing structural problems. The data part is also missing which makes the article to seem more problematic. I wish that the authors will significantly improve their article and submit a fresh modified version. Here I listed just some of the problems that were more obvious, however, many other problems exist that are not listed here.

  • Line 19: “Has become” Please use proper English.
  • Line 25: “the small size 24 of drones- >” small sized of drones‘’
  • Line 26: “crowded backgrounds” is this really a problem when detecting an object in sky? It will be if you consider clouds as crowd.
  • Line 26: All abbreviations should be expanded upon their first use.
  • Line 30: All abbreviations should be expanded upon their first use.
  • Line 26-30: What accuracy you are talking here? Is it training or validation?
  • Line 30: ”a 4% improvement” -> A slight improvement of 4 percent.
  • Line 38: “aircraft” -> flying vehicles.
  • Line 42-43: Detecting UAVs is not a task of a surveillance system. You should be careful about the rerms that have specific usages.
  • Line 48: This reference is by no means necessary to be addressed here. Detecting an object from background is a well-known concept. (inappropriate self-citation)
  • Line 49-51: You are talking here about classifying drones and then identifying their subclass. This sentence is very vague in my opinion. It is generally a good idea to firstly detect a drone from sky and then talk about their subtype, however, it needs to be stated in a better way. In general, I advise you to fully revise your article, since there are many parts that could mislead the reader.
  • Line 55, a space is required before parenthesis.
  • Line 55-59: These reasons are not acceptable to me. The only major benefit that I see is the cost. Mounting is not a significant issue at all.
  • Line 64-66. Speed improvement is not only based on invention of deep neural networks. It is also a result of employing better hardware such as graphical and tensor processing units or CPUs.
  • Line 68-70, the scientific soundness of this statement is very weak in my opinion.
  • Line 72: “More features are extracted”. What do you mean by feature? Do you mean the number of trainable parameters are more? This is also a weak statement.
  • Line 77: A dot is redundant after the parenthesis.
  • Line 77-79: I understand it this way that the whole image is processed that leads to better performance. This should be reverse. I expect to get a faster result when a candidate area is firstly selected.
  • Line 79-81. Very ambiguous statement about YOLO. Please improve. I suggest that you meticulously read the YOLO article to get a clear understanding about how it works.
  • “extracting features and object detection” -> “extracting features and detecting objects”.
  • Line 85. Very poor English sentence. “of detection and 85 recognition” should be removed in my opinion.
  • Line 86: What do you mean the network is not optimized? Do you mean that that there is a level of optimization that this network lacks? or maybe you mean that it is not complex enough to recognize some types of objects. Very vague statement.
  • Line 95: Why there is no numbering here?
  • Line 98, two sentences were inappropriately connected.
  • Line 102: No numbering! all the sub chapters should be numbered.
  • Line 103: For human or CV algorithm or both?
  • Line 108: no numbering.
  • Line 114: no numbering.
  • Line 120: no numbering.
  • Line 128: a weak conclusion.
  • Line 138. The citation doesn’t follow the style. A punctuation is missed.
  • Line 138-139: A bad start for literature review. The statement is very vague.
  • Why there is almost no CNN literature review?
  • Line 219: “Developed steps”?
  • Line 221: “is prepared”->”was prepared”. Please correct on many other occations. Past tense should be used when some task was performed in past.
  • Line 228: Very vague sentence.
  • Line 256-258: Wrong statement. A background detection algorithm could be initially used to ease this step.
  • Line 260: Marking a UAV with a polygon is much harder in comparison to marking it with a rectangle. Is your goal to put label on every pixel? If this is the case, then you are dealing with a semantic segmentation problem which in nature is different from an image classification problem at patch level. This should be clarified in the introduction of your method.
  • Line 271: What sort of modification you did on Yolo? What are the benefits?
  • Line 273: Weak English. Wrong statement.
  • Line 396: There is no need to put formulas for famous classification evaluation metrics. A proper reference could be cited e.g.
  • Where is the data part? How many images you were manually labelled?
  • Line 485: “is changed”->”was changed”
  • Line 491-492: The statement is wrong.
  • Line 505-506. You don’t need to describe the use of confusion matrix here.
  • Line 613. You are repeating literature review here, however, the aim should be to compare your main funding with other people. The structure of the discussion is not right in my opinion.
  • The contribution of each author is not stated at the end of the article.

Author Response

Response to reviewers' comments

 The Drones journal

Ms. Rocksy Zhang:

Thank you very much for reviewing our manuscript. We also greatly appreciate the reviewers for their complimentary comments and suggestions. We have carried out the experiments that the reviewers suggested and revised the manuscript accordingly. We hope that you find the responses satisfactory and that the manuscript is now acceptable for publication.

Reviewer 2

  1. Line 19: “Has become” Please use proper English.

Response: Thank you for your constructive comments. this point has been corrected in lines 21 to 23.

--------------------------------------------------------------------------------------------------------------------------

  1. Line 25: “the small size of drones- >” small sized of drones‘’

Response: This means "small drones" which has been corrected in line 27.

  1. Line 26: “crowded backgrounds” is this really a problem when detecting an object in sky? It will be if you consider clouds as crowd.

Response:  Clouds are one of the categories of a crowded background in the sky. We are considering everything, including dust, fog, and fire. For the audience's better understanding, the crowded background is described in detail in line 128 of section 1.1.2

--------------------------------------------------------------------------------------------------------------------------

  1. Line 26: All abbreviations should be expanded upon their first use.
  2. Line 30: All abbreviations should be expanded upon their first use.

Response: These points have been corrected in lines 28 and 33.

--------------------------------------------------------------------------------------------------------------------------

  1. Line 26-30: What accuracy you are talking here? Is it training or validation?

Response: The reported results refer to the test phase of the network. This point has been corrected in line 32.

-------------------------------------------------------------------------------------------------------------------------------

  1. Line 30: ”a 4% improvement” -> A slight improvement of 4 percent.

Response: This point has been corrected in line 34.

-------------------------------------------------------------------------------------------------------------------------------

  1. Line 38: “aircraft” -> flying vehicles.

Response: This point has been corrected in line 48.

-------------------------------------------------------------------------------------------------------------------------------

  1. Line 42-43: Detecting UAVs is not a task of a surveillance system. You should be careful about the terms that have specific usages.

Response: Regarding this comment, we corrected this sentence in lines 50 to 52.

-------------------------------------------------------------------------------------------------------------------------------

  1. Line 48: This reference is by no means necessary to be addressed here. Detecting an object from background is a well-known concept. (inappropriate self-citation)

Response: The point has been corrected.

-------------------------------------------------------------------------------------------------------------------------------

  1. Line 49-51: You are talking here about classifying drones and then identifying their subclass. This sentence is very vague in my opinion. It is generally a good idea to firstly detect a drone from sky and then talk about their subtype, however, it needs to be stated in a better way. In general, I advise you to fully revise your article, since there are many parts that could mislead the reader.

Response: Conventional automatic target recognition (ATR) methods first verify the presence or absence of drones in the scene and then perform drone type recognition. Unlike conventional drone detection technologies, the nature of deep learning networks is to perform drone detection and recognition simultaneously. By dividing the input data into multiple classes, these networks determine the presence, absence, and type of the drone class. In this paper, the input data is firstly divided into 5 classes and the deep learning network is trained with this data to then each of these classes is recognized in the testing phase. For example, if a multirotor is observed in the sky, its existence and the type of the class are determined simultaneously.

According to this comment, these items are corrected in lines 59 to 61, and 73 to 76.

-----------------------------------------------------------------------------------------------------------------------------

  1. Line 55, a space is required before parenthesis.

Response: This point is corrected throughout the article in similar cases.

-------------------------------------------------------------------------------------------------------------------------------

  1. Line 55-59: These reasons are not acceptable to me. The only major benefit that I see is the cost. Mounting is not a significant issue at all.

Response: This part of the article assumes the situation where drones can be detected in the sky by another drone. In this case, there is the problem of integrating the required sensors with the drone. However, due to the ambiguity and the fact that our platform is terrestrial, this part has been removed.

-----------------------------------------------------------------------------------------------------------------------------

  1. Line 64-66. Speed improvement is not only based on invention of deep neural networks. It is also a result of employing better hardware such as graphical and tensor processing units or CPUs.

Response: Thanks for the constructive comment. This point is corrected in lines 70 to 72.

-----------------------------------------------------------------------------------------------------------------------------

  1. Line 68-70, the scientific soundness of this statement is very weak in my opinion.

Response: This part is corrected in lines 83 and 84.

-----------------------------------------------------------------------------------------------------------------------------

  1. Line 72: “More features are extracted”. What do you mean by feature? Do you mean the number of trainable parameters are more? This is also a weak statement.

Response: By "features" in this article we mean descriptors that describe features such as color, texture, and shape of the object in the image. The use of the word "features" is more common than the "descriptor" word in the context of a Deep Learning network. Therefore, we have used this word in this article.

The extracted properties are essentially descriptors of objects, and as their number increases, object detection are performed with greater accuracy.

The increase in extracted features is due to changes in the architecture and layers of the deep learning network. This adds several parameters to the network training parameters. However, due to the importance of accuracy in deep learning problems, the increase in the number of parameter is acceptable in comparison to the improvement of the accuracy.

According to the comment, these points are addressed in lines 88 to 90.

-----------------------------------------------------------------------------------------------------------------------------

  1. Line 77: A dot is redundant after the parenthesis.

Response: This point is corrected.

-----------------------------------------------------------------------------------------------------------------------------

  1. Line 77-79: I understand it this way that the whole image is processed that leads to better performance. This should be reverse. I expect to get a faster result when a candidate area is firstly selected.

Response: The speed of object detection in a network depends on the network architecture in addition to the size of input. Region-based object detection methods have a different and more complex architecture than YOLO. This fact slows them down compared to YOLO network.

According to this comment, these points are corrected in lines 95 and 96.

-----------------------------------------------------------------------------------------------------------------------------

  1. Line 79-81. Very ambiguous statement about YOLO. Please improve. I suggest that you meticulously read the YOLO article to get a clear understanding about how it works.

Response: According to this comment, these points are considered in lines 96 to 99.

-----------------------------------------------------------------------------------------------------------------------------

  1. Line 85. Very poor English sentence. “of detection and recognition” should be removed in my opinion.

Response: This point is corrected.

-----------------------------------------------------------------------------------------------------------------------------

  1. Line 86: What do you mean the network is not optimized? Do you mean that that there is a level of optimization that this network lacks? or maybe you mean that it is not complex enough to recognize some types of objects. Very vague statement.

Response:. By "optimize" we mean that the basic YOLOv4 network is less capable of addressing the challenges described in Section 1.1. It is better to use the phrase "lack of acceptable capabilities" instead of the word "optimize".

According to this comment, the point is addressed in lines 102 to 105.

-------------------------------------------------------------------------------------------------------------------------------

  1. Line 95: Why there is no numbering here?
  2. Line 102: No numbering! all the sub chapters should be numbered.
  3. Line 108: no numbering.
  4. Line 114: no numbering.
  5. Line 120: no numbering.

Response: These points are corrected in lines 111,117,125,132,139.

-------------------------------------------------------------------------------------------------------------------------------

  1. Line 98, two sentences were inappropriately connected.

Response: According to this comment, lines 114 and 115 are revised.

-------------------------------------------------------------------------------------------------------------------------------

  1. Line 103: For human or CV algorithm or both?

Response: Due to the similarity of the physical structure of drones with birds, humans face difficulty in distinguishing them from each other. According to this comment, the point is addressed in lines 118 and 119.

-------------------------------------------------------------------------------------------------------------------------------

  1. Line 128: a weak conclusion.

Response: This part is revised in lines 149 to 151.

-------------------------------------------------------------------------------------------------------------------------------

  1. Line 138-139: A bad start for literature review. The statement is very vague.

Response: This part is rewritten in lines 154 to 156.

-------------------------------------------------------------------------------------------------------------------------------

  1. Why there is almost no CNN literature review?

Response: Thanks for your comment. We have answered your question in two parts of the article:

First, we explained the architecture of the convolutional neural network in the introduction in lines 76 to 83. Then, in the literature review section, in lines 166 to 173, we discussed an article that detects a drone with this network.

-------------------------------------------------------------------------------------------------------------------------------

  1. Line 219: “Developed steps”?

Response: This point is corrected in line 237.

-------------------------------------------------------------------------------------------------------------------------------

  1. Line 221: “is prepared”->”was prepared”. Please correct on many other occations. Past tense should be used when some task was performed in past.

Response: These points are corrected.

-------------------------------------------------------------------------------------------------------------------------------

  1. Line 228: Very vague sentence.

Response: This point is corrected.

-------------------------------------------------------------------------------------------------------------------------------

  1. Line 256-258: Wrong statement. A background detection algorithm could be initially used to ease this step.

Response: Due to the time-consuming process of image labeling, we will definitely use your idea in future research. We also added this point in the conclusion section as future work in lines 699 and 700.

-------------------------------------------------------------------------------------------------------------------------------

  1. Line 260: Marking a UAV with a polygon is much harder in comparison to marking it with a rectangle. Is your goal to put a label on every pixel? If this is the case, then you are dealing with a semantic segmentation problem which in nature is different from an image classification problem at patch level. This should be clarified in the introduction of your method.

Response: Yes, labeling is done for each pixel. Fewer unrelated pixels in the input label will result in less error detection and recognition. For this reason, drawing polygons instead of rectangles is a good operation, since it removes many unrelated pixels. We clarified this point in the “input preparation” section, lines 262 to 266.

-------------------------------------------------------------------------------------------------------------------------------

  1. Line 271: What sort of modification you did on Yolo? What are the benefits?

Response: In this network, the addition of three convolutional layers after the backbone section increased the extracted features, resulting in higher recognition accuracy and better overcoming to the existing challenges. In addition, adding a convolutional layer with a size of 3×3 and another with a size of 1×1 improved the detection of small objects and reduced the volume of network computation.

We have added the explanation this in a separate section called "The Modified YOLOv4 Deep Learning Network Architecture" in lines 346 to 367.

-------------------------------------------------------------------------------------------------------------------------------

  1. Line 273: Weak English. Wrong statement.

Response This part is rewritten.

-------------------------------------------------------------------------------------------------------------------------------

  1. Line 396: There is no need to put formulas for famous classification evaluation metrics. A proper reference could be cited e.g.

Response: Yes, this point has been corrected and proper references are added.

-------------------------------------------------------------------------------------------------------------------------------

  1. Where is the data part? How many images you were manually labelled?

Response: In this regards, we have added section 4.1 as “data preparation”.

-------------------------------------------------------------------------------------------------------------------------------

  1. Line 485: “is changed”->”was changed”

Response: This point has been corrected throughout the article in similar cases.

-------------------------------------------------------------------------------------------------------------------------------

  1. Line 491-492: The statement is wrong.

Response: This point has been corrected in line 485.

-------------------------------------------------------------------------------------------------------------------------------

  1. Line 505-506. You don’t need to describe the use of confusion matrix here.

Response: This point has been corrected.

-------------------------------------------------------------------------------------------------------------------------------

  1. Line 613. You are repeating literature review here, however, the aim should be to compare your main funding with other people. The structure of the discussion is not right in my opinion.

Response: Thanks for your comment. The discussion chapter is rewritten.

-------------------------------------------------------------------------------------------------------------------------------

  1. The contribution of each author is not stated at the end of the article.

Response: This section is added at the end of the article.

-------------------------------------------------------------------------------------------------------------------------------

  • In addition to the above corrections, the text has been revised and many style problems and vague sentences have been corrected.

-------------------------------------------------------------------------------------------------------------------------------

-------------------------------------------------------------------------------------------------------------------------------

Round 2

Reviewer 2 Report

Most of the issues that I have pointed out previously have been addressed.

Reviewer 3 Report

The changes are still unsatisfying to me. The main issue about emphasizing the novelty of this work is still missing. Therefore I reject this paper.